# Variables Associated with False-Positive PSA Results: A Cohort Study with Real-World Data

**DOI:** 10.3390/cancers15010261

**Published:** 2022-12-30

**Authors:** Blanca Lumbreras, Lucy Anne Parker, Juan Pablo Caballero-Romeu, Luis Gómez-Pérez, Marta Puig-García, Maite López-Garrigós, Nuria García, Ildefonso Hernández-Aguado

**Affiliations:** 1Department of Public Health, University Miguel Hernández de Elche, 03550 Alicante, Spain; 2CIBER de Epidemiología y Salud Pública (CIBERESP), 28029 Madrid, Spain; 3Department of Urology, University General Hospital of Alicante, 03010 Alicante, Spain; 4Alicante Institute for Health and Biomedical Research (ISABIAL), 03010 Alicante, Spain; 5Urology Department, General University Hospital of Elche, 03203 Elche, Spain; 6Pathology and Surgery Department, Miguel Hernández University of Elche, 03550 Alicante, Spain; 7Clinical Laboratory Department, University Hospital of San Juan de Alicante, Sant Joan d’Alacant, 03550 Alicante, Spain

**Keywords:** prostate-specific antigen, prostate cancer, false-positive results, real world-data

## Abstract

**Simple Summary:**

Controversy exists regarding prostate cancer (PC) screening using the prostate-specific antigen (PSA) test. It may reduce PC mortality risk but is associated with false-positive results. We aimed to evaluate the incidence of false-positive and false-negative results in a general clinical setting and the associated variables. We found a high rate of false-positive results (46.6%), resulting in a positive predictive value of 12.7%. Patients also showed a low rate of false-negative results (3.7%) with a negative predictive value of 99.5%. Age, alcohol intake, and having a urinary tract infection were associated with a higher probability of false-positive results; having diabetes mellitus type II was associated with a lower rate of false-positive results. This study showed a higher rate of false-positive results in clinical practice than in previous clinical trials, mainly in patients over 60 years.

**Abstract:**

(1) Background: There are no real-world data evaluating the incidence of false-positive results. We analyzed the clinical and analytical factors associated with the presence of false-positive results in PSA determinations in practice. (2) Methods: A prospective cohort study of patients with a PSA test was performed in clinical practice. We followed the patients by reviewing their medical records for 2 years or until the diagnosis of PCa was reached, whichever came first. (3) Results: False-positive PSA rate was 46.8% (95% CI 44.2–49.2%) and false-negative PSA rate was 2.8% (95% CI 2–3.5%). Patients aged 61–70 years and those over 70 years were more likely to have a false-positive result than those under 45 years (aOR 2.83, 95% CI 1.06–7.55, *p* = 0.038, and aOR 4.62, 95% CI 1.75–12.22, *p* = 0.002, respectively). Patients with urinary tract infection were more likely to have a false-positive result (aOR 8.42, 95% CI 2.42–29.34, *p* = 0.001). Patients with diabetes mellitus were less likely to have a false-positive result (aOR 0.63, 95% CI 0.41–0.98, *p* = 0.038); (4) Conclusions: This study has generated relevant information that could be very useful for shared decision making in clinical practice.

## 1. Introduction

In recent years there has been a debate regarding the benefits and harms of PSA-based screening for prostate cancer (PCa) [1,2]. A systematic review of randomized controlled trials published in 2013 showed that PSA-based PCa screening did not significantly decrease PCa-specific mortality. In addition, harms related to the presence of overdiagnosis, overtreatment, and false-positive results were frequent [3]. Concerning the presence of false-positive results, previous evidence has also shown variations in PSA levels according to age [4], although there is no evidence of their implications in practice. A recent systematic review [5] showed that diabetes was significantly associated with lower PSA levels among asymptomatic men. Though, differences were small and unlikely to influence PCa detection. Similarly, treatments such as statins [6], metformin [7], or treatment for benign prostatic hypertrophy [8] can also affect PSA results leading to false-positive or -negative results. According to previous studies carried out in clinical trials [9], 10–12% of men undergoing regular PSA testing will experience a false-positive result. False-positive results can have a major impact on the clinical management and outcome of the patient, mainly due to possible adverse effects related to the diagnostic process (biopsy, surgery, and treatment). Biopsies, for instance, can cause infections as well as very serious complications such as urinary incontinence and sexual dysfunction [10].

These data led some international organizations, such as the US Preventive Services Task Force (UPSTF) in 2012, to establish recommendations against PSA-based screening for PCa [11]. In 2018, based on longer follow-up data from large screening trials, the UPSTF guideline was updated [12]. They stated that men aged 55–69 years should be informed about the benefits and harms of PCa screening with a grade C recommendation (the USPSTF recommends selectively offering or providing this service to individual patients based on professional judgment and patient preferences), while grade D (the USPSTF recommends against the service) was assigned to screening for men over 70 years. Aiming to reduce the harms related to false-positive results and overdiagnosis, the European Association of Urology guidelines have been recently updated [13] and recommend offering an individualized risk-adapted approach for the early detection of PCa to well-informed men older than 50 years old with at least 15 years of life expectancy.

However, most of these recommendations are based on randomized clinical trials, which include populations with different characteristics from those patients in clinical practice [14] (clinical trials tend to include healthier and younger patients and often represent a highly selected population of patients). Indeed, PSA levels in patients attending urology services are higher than those patients of the same age who participate in screening programs [15]. Therefore, both the baseline levels of the marker and the determinants that may affect a false-negative or false-positive result should be determined for each setting.

The risk-adapted approach for the early detection of PCa, as the European Association proposes, should be developed along with the evaluation of the probability of a patient having a false-positive or -negative result according to his individual characteristics. Nevertheless, at present, there are no real-world data evaluating the incidence of false-positive and -negative results in practice and the variables associated with them.

We aimed to evaluate the incidence of false-positive and false-negative results in a general clinical setting, including patients undergoing opportunistic screening or with symptoms suggestive of disease, and the variables associated.

## 2. Materials and Methods

This study was carried out and reported according to the Strengthening the Reporting of Observational Studies in Epidemiology (STROBE) Statement [16].

### 2.1. Study Design

A prospective observational cohort study of patients with a PSA determination for the early detection of PCa or in the presence of prostatic symptoms in general practice. The study protocol is registered at https://clinicaltrials.gov (ClinicalTrials.gov Identifier: NCT03978299, accessed on the 11 June 2019) and has been previously published [17].

### 2.2. Setting

The target population of the study were the residents of the catchment area of the two participating Health Departments 17 and 19, in the Valencian Community (these include 20 primary health centers and 2 hospitals: General University Hospital of Sant Joan d’Alacant and General University Hospital of Alicante, respectively). These are referral hospitals for all individuals living in their catchment areas and belong to the National Health System (NHS). Most of the Spanish population uses it as the main medical service (the publicly funded insurance scheme covers 98.5% of the Spanish population).

### 2.3. Study Population

We included men over 18 years with a PSA determination requested in a routine health examination from January to July 2018. Patients with a previous diagnosis of PCa or those who were being monitored for previous high PSA values were excluded.

We randomly selected a sample of patients with positive PSA results from within our cohort (total PSA value >10 mg/L or a total PSA between 4 and 10 mg/L if the value of the free PSA/total PSA fraction is <25% in at least two determinations) and a sample of patients with negative results (total PSA value is < 4 mg/L or a total PSA between 4 and 10 mg/L if the value of the free PSA/total PSA fraction is >25%) among a consecutive cohort of individuals undergoing PSA testing that have been described previously [18]. This previous study aimed to evaluate the potential non-compliance of PSA testing with current guidelines in general practice. PSA determinations are performed centrally in the laboratories of these two hospitals using the same protocol. A blood sample is extracted, and after centrifugation (15 min), the PSA level is determined in the serum using the chemiluminescent assay technique (the analyte is stable for 4 days at 2–8 ºC). The detection limit of the technique is 0.05 ng/mL.

### 2.4. Study Size and Recruitment Procedure

The predictive positive value of PSA is estimated at 21%, and the predictive negative value at 91%, according to previous evidence [19]. Given that we included symptomatic and asymptomatic patients, we estimated a prevalence of PCa in our study higher than 5%. We calculated that we needed to include at least 665 patients with a negative PSA result and 690 PSA-positive patients (for a precision of 2% with 95% CIs). Considering the potential loss during follow-up, we increased this sample by 20%.

### 2.5. Data Collection

Both hospitals have digital medical records from which data were extracted.

We recorded the following data from medical records: demographic characteristics (age) and clinical characteristics (patient who had the PSA test as part of opportunistic screening or due to the presence of symptoms suggestive of disease), toxic habits (smoking habit and alcohol consumption), family history of PCa, any pharmacological treatment prescribed and specifically treatment with statins, metformin, treatment for BPH, diuretics and ASA, PSA values, anthropometric measures, and other comorbidities.

### 2.6. Cohort Follow-Up

We followed both cohorts (positive and negative PSA results) for 2 years by reviewing their medical records (every 3 months for patients with a positive PSA result and annually for patients with a negative result) until the diagnosis of PCa or end of follow-up, whichever came first. We also collected results from digital rectal exams and biopsies. For those patients with a diagnosis of PCa, data regarding the Gleason score and the International Society of Urological Pathology grade were also collected.

### 2.7. Outcomes

A false-positive result was defined if the determination of serum PSA was positive and the result of digital rectal examination(s) and/or subsequent biopsy or biopsies were negative, according to the recommendations of the European Association of Urology [20]. A false-negative result was defined if the determination of serum PSA was negative and the patient was diagnosed with PCa within 2 years.

### 2.8. Statistical Analysis

We used descriptive statistics to summarize the population and calculated the proportion of false-positive and false-negative results for the diagnosis of PCa and the associated variables (Chi-squared test). To analyze clinical and analytical factors associated with the presence of false-positive results in PSA determinations, we calculated odds ratios and their 95% confidence interval with logistic regression. Independent variables included were urinary tract infection, diabetes mellitus type II, hyperlipidemia, age, alcohol consumption, family history of prostate cancer, and prescription of any pharmacological treatment.

The analysis was performed using the Stata IC 15 program. All *p* values and CIs were two-sided, 95%.

## 3. Results

### 3.1. Description of The Patients Included in Both Cohorts and Accuracy of the PSA Test

Of the 1664 patients included in the study, 833 (51%) had a positive PSA result, and 831 (49%) had a negative PSA result. Out of 833 patients with a positive PSA result, 106 (97.2%) had a diagnosis of PCa; of the 831 patients with a negative PSA result, 3 (2.8%) had a diagnosis of PCa. The positive predictive value (PPV) of the PSA determination was 12.7% (95% CI 10.4–15%), and the negative predictive value (NPV) was 99.6% (95% CI 99.2–100%). The false-positive PSA rate was 46.8% (95% CI 44.2–49.2%), and the false-negative PSA rate was 2.8% (95% CI 2–3.5%).

Median follow-up was 19.51 months (IQR 14.39–22.24), and median follow-up until the patient was diagnosed with PCa was 5 months (IQR 11–16).

Of the 1664 patients included in the study, 33 (1.9%) had a family history of PCa and 2 of them (6.1%) developed PCa, 169 (10.1%) had no family history of prostate cancer, and 19 (11.2%) developed cancer; however, no data on family history of PCa were available for 1462 patients (87.9%).

### 3.2. False-Positive Rate and Associated Sociodemographic and Clinical Characteristics

Table 1 shows the distribution of the number of patients with PCa and the false-positive PSA rates, according to the different sociodemographic and clinical variables, for symptomatic and asymptomatic patients.

Overall, there were differences in the false-positive rate according to patients’ age: the rate increased from 26.3% in patients younger than 45 years to 55.2% in patients over 70 years (*p* < 0.001). In asymptomatic patients, a difference in the false-positive rate according to patients’ age was also found (*p* < 0.001).

Most patients had a previous PSA (1347/1665; 80.9%), and they were less likely to have a false-positive result than patients who had a PSA test for the first time (45.1% vs. 53.8%, *p* = 0.008). In asymptomatic patients, a difference in the false-positive rate according to having a previous PSA was also found (*p* = 0.023).

Patients who never had drunk alcohol (386/1665; 23.2%) were less likely to have a false-positive result (46.2%) than those who were current (56.6%) or ex-drinkers (58.5%) (*p* = 0.014). In asymptomatic patients, a difference in the false-positive rate according to alcohol intake was also found (*p* = 0.004).

### 3.3. False-Positive Rate Associated with The Presence of Comorbidities

Table 2 shows the distribution of the number of patients with PCa and the false-positive PSA rates, according to the different patients’ comorbidities, for symptomatic and asymptomatic patients.

Patients with hyperlipidemia were less likely to have a false-positive result (40% vs. 48.2%, *p* = 0.016). In asymptomatic patients, a difference in the false-positive rate according to the presence of hyperlipidemia was also found (*p* = 0.012). The presence of diabetes mellitus type II was also associated with a lower probability of having a false-positive result (36.5% vs. 48.4%, *p* = 0.002). Both symptomatic and asymptomatic patients with diabetes mellitus type II had a lower probability of having a false positive than those without diabetes (*p* = 0.008 and *p* = 0.019, respectively). In contrast, the presence of urinary tract infection was associated with a higher probability of false-positive results (80.4% vs. 45.8%, *p* < 0.001) for both symptomatic (*p* < 0.001) and asymptomatic (*p* < 0.001) patients.

### 3.4. False-Positive Rate Associated with the Prescription of Medication

Patients with a prescribed pharmacological treatment were less likely to have a false-positive result (44.8% vs. 54.1%, *p* = 0.003). In asymptomatic patients, a difference in the false-positive rate according to the prescription of pharmacological treatment was also found (*p* = 0.012) (Table 3). However, there were no differences in the false-positive rate according to the specific pharmacological treatments.

In multivariable analysis, patients with urinary tract infections were more likely to have a false-positive result (aOR 8.42, 95% CI 2.42–29.34, *p* = 0.001). This was also the case for patients who were current drinkers in comparison with never drinkers (aOR 1.48, 95% CI 1.10–2.02, *p* = 0.011). Patients with diabetes mellitus were less likely to have a false-positive result (aOR 0.63, 95% CI 0.41–0.98, *p* = 0.038). Moreover, patients aged 61–70 years and those over 70 years were more likely to have a false-positive result than those under 45 years (aOR 2.83, 95% CI 1.06–7.55, *p* = 0.038, and aOR 4.62, 95% CI 1.75–12.22, *p* = 0.002, respectively).

### 3.5. Description of the PCa Cases and their Main Characteristics

Of the 109 patients with PCa, 94 (86.2%) were able to be classified according to the Gleason score: 31 (33%) were classified as Gleason 6, 36 (38.3%) as Gleason 7, and 27 (28.7%) as Gleason 8–10. There were statistically significant differences in the median PSA value for each Gleason category (*p* = 0.006) (Table 4).

According to the International Society of Urological Pathology grade, 43 (39.4%) patients were classified as: grade 1, 22 (51.2%); grade 2, 7 (16.3%); grade 3, 10 (23.3%); grade 4, 5 (11.6%), and grade 5, 1 (2.3%). There were also differences in the median PSA values according to this classification: from a PSA median of 4 (IQR 1.67–6.04) in patients with grade 1 to a PSA median of 10.54 in patients with grade 5 (*p* < 0.001).

Of the 1665 patients included in the study, 303 (18.2%) had symptoms suggestive of prostate disease, and 17 (5.6%) developed PCa; of the 1361 (81.8%) patients without symptoms, 92 (6.8%) developed PCa. Cancer patients who had been symptomatic at the time of the PSA test had a Gleason score of 6–7 (7, 46.7%) and a Gleason score of 8–10 (8, 53.3%). At the same time, those who had been asymptomatic had a Gleason score < 6 (6, 7.7%), a Gleason score of 6–7 (54, 68.4%), and a Gleason score of 8–10 (19, 24.4%), *p* = 0.059.

In patients who did not have a PCa diagnosis, there were differences in PSA level according to age (*p* = 0.001); these differences were not found in patients with PCa (*p* = 0.072) (Figure 1).

In patients without PCa, having hyperlipidemia, diabetes, or a prescribed pharmacological treatment was associated with lower PSA levels (*p* = 0.009, *p* = 0.001, and *p* < 0.001, respectively); having an infection of the urinary tract was associated with higher PSA levels (<0.001). These differences were not observed in patients with PCa (Table 5).

## 4. Discussion

We found a high rate of false-positive results both in symptomatic (47.9%) and asymptomatic patients (46.6%), resulting in a PPV of 12.7% in clinical practice. Patients in this study also showed a low rate of false-negative results (3.7%) with an NPV of 99.5%.

This is the first study based on real-world data that includes both symptomatic and asymptomatic patients, and thus, it is difficult to compare our results with similar studies. However, it was possible to compare our results with previous evidence based on clinical trials, and we found a greater rate of false-positive results. Results from randomized clinical trials showed the PPV of an elevated PSA in the 20–30% range [21], higher than the PPV shown in our study. Furthermore, according to a recent meta-analysis of clinical trials, 4–19% of all screening participants had a false-positive screening result. In our study, PSA values higher than 4.0 ng/mL were used as an indication for biopsy among men of all ages. Given that most of the available clinical trials used a 3.0 ng/mL threshold, we consider that false-positive results in these clinical trials should be higher. These differences in the false-positive rates could reflect the different populations included in clinical practice (older patients and with several comorbidities such as UTI) in contrast with clinical trials.

The level of PSA in the blood increases with age by about 3.2% per year [22], leading to an increase in the false-positive rate in the PSA test. Therefore, according to previous results, the use of this threshold for recommending prostate biopsy is not adequate for men of all ages [23]. Nevertheless, the usefulness of age-adjusted PSA thresholds is controversial due to the risk of missing a high proportion of clinically significant cancers in older men. In our study, there were no differences in the PSA values in those patients with PCa according to age, while there were differences in patients without PCa. These results suggest that an age-adjusted PSA threshold for biopsy may be of limited use despite the increase in the false-positive results in the PSA test with patients’ age. In addition, given the low incidence of false-negative results and the PSA levels of the patients with PCa, the value of 4 ng/mL seems a reasonable threshold. At the same time, prostate cancer overdiagnosis (cases that would not have caused clinical consequences during a man’s lifetime if left untreated) has a strong relationship to age [24]. A relevant percentage of the prostate cancer cases detected in asymptomatic patients included in the study could be overdiagnosed cases. Hence, restricting screening in patients older than 70 years old, as most available guidelines recommend, could importantly reduce overdiagnosis.

We have shown that several factors are associated with a higher probability of false-positive results: age, alcohol consumption, and having a urinary tract infection. In contrast, having diabetes mellitus type II was associated with a lower rate of false-positive results. Differences related to the presence of diabetes mellitus, urinary tract infection, and age were only observed in men who had no urinary symptomatology when the PSA test was ordered. This could be explained by the lack of precision in patients with symptoms or that the presence of symptoms acted as an effect modifier. Patients with hyperlipidemia and diabetes showed lower PSA levels than those without these pathologies. A recent systematic review showed that diabetes was significantly associated with lower PSA levels among asymptomatic men older than 60 years, leading to a lower probability of having a false-positive PSA result [5]. The implication is that the predictive value of a positive test is higher among patients with diabetes or hyperlipidemia. One could consider that the lower range of PSA values identified in diabetics compared to non-diabetics and in patients with hyperlipidemia compared to those without hyperlipidemia justifies a different cut-off point when considering a positive PSA result in this population. However, given that there were no differences in PSA values among patients with PCa, it is unlikely that using a specific cut-off would improve the accuracy of PSA screening.

In any case, this could be a useful observation for clinicians when interpreting PSA results in routine practice.

Several studies have developed risk prediction models to avoid false-positive results based on biomarkers and sociodemographic and clinical variables [25]. However, none of these models have been validated in clinical practice with positive results. The STHLM3 study [26] developed and validated a model combining plasma protein biomarkers, genetic polymorphisms, and clinical variables. It demonstrated a significant improvement in the specificity of PCa screening with the same sensitivity as PSA testing. This study included men in a specific age range, 50–69 years, whereas those undergoing PSA testing in clinical practice came from a wider age range. Given the differences between the populations included in the clinical trials and clinical practice, we consider that further research should be conducted in clinical practice to evaluate the addition of other clinical and genetic factors. In contrast with previous studies, we did not show a relationship between BMI and PSA levels. A previous study of Korean men showed a significant inverse relationship with BMI in overweight and obese men aged 40–59 years. However, there was no relationship between serum PSA and BMI in men older than 60 years [27]. In previous studies aimed at developing a predictive model for PCa diagnosis based on marching learning, the inclusion of PSA level and patient age increased the accuracy of the model, while BMI had only a minimal effect [28].

In our study, there were no differences in false-positive and false-negative PSA results according to whether the patient had a pharmacological prescription in univariate analysis but not in multivariable analysis. There were also no differences in PSA levels according to the type of treatment. In a previous longitudinal study [6], PSA levels declined to a statistically significant extent after the initiation of statin treatment, which can complicate cancer detection. In another study [7], metformin was found to have a dose-dependent inverse relationship with serum PSA levels. A previous clinical trial found that finasteride decreased PSA levels [29] and concluded that new prostate biomarkers should be interpreted with caution in patients receiving these treatments. However, another research [30] concluded that although both dutasteride and finasteride reduced PSA levels, dutasteride should not be considered equivalent to finasteride in the reduction rate of PSA. In our study, the absence of statistical significance when analyzing treatment type could perhaps be explained by the limited statistical power. Moreover, nearly 80% of patients had a current prescription for a pharmacological treatment (and most of them had two or more concomitant treatments). Hence, it was difficult to analyze the impact of an individual treatment on the false-positive rate in the univariate analysis.

There were also no differences in the probability of having a PCa according to the presence of symptoms (5.6% vs. 6.8%), and the false-positive rate was similar for both symptomatic and asymptomatic patients. A recent systematic review of randomized clinical trials [1] showed that in men without symptoms benefits of PSA-based PCa screening did not outweigh the harms, but our study showed that the presence of symptoms does not seem to have a relevant impact on the PCa diagnosis.

A population study in Norway showed how opportunistic PSA testing substantially increased the incidence of localized and regional prostate cancers among men aged 50–74 years [31]. In our study, there were no statistically significant differences according to the Gleason score distribution between symptomatic and asymptomatic patients, but symptomatic patients were more likely to have a Gleason score of 8–10 than asymptomatic patients. However, given the follow-up period of the study, we were unable to assess the impact on long-term mortality.

Recently, the European Association of Urology, the European Association of Nuclear Medicine-European Society for Radiotherapy, the Oncology-European Society of Urogenital Radiology-International, and the Society of Geriatric Oncology have published the guidelines on screening, diagnosis, and local treatment of clinically localized PCa [20]. As a screening strategy, they advise a risk-adapted screening to identify men who may develop PCa, from age 50 based on individualized life expectancy. However, this risk-adapted screening should be offered to men at increased risk from the age of 45 and to carriers of breast cancer susceptibility gene (BRCA) mutations. The use of multiparametric MRI is also recommended to avoid unnecessary biopsies. A retrospective study found that bi-parametric prostate MRI was a powerful tool in the detection of clinically significant PCa, but PSA density did not appear to significantly improve its diagnostic performance [32]. In addition, a recent clinical trial found that MRI-directed targeted biopsy for screening and early detection in persons with elevated PSA levels reduced the risk of overdiagnosis [33].

International guidelines strongly recommend incorporating shared decision making into PSA-based prostate cancer screening [20]. Patients’ knowledge and attitudes toward PCa screening are decisive factors in the adoption of opportunistic screening [34], and there are variations across the world [35]. In Italy, for example, knowledge about PCa screening amongst male subjects is quite high, although knowledge of PCa risk factors, mainly both genetic/hereditary, is low, and the practice of DRE is underutilized [36]. Considering the new recommendations published by European Commission and Urologist Associations [20], education interventions should be implemented to allow patients to participate in shared decision-making regarding PSA opportunistic screening.

This study has several limitations. Firstly, we followed our patients until they were diagnosed with PCa or for 2 years (whichever came first), and therefore, this period was too short to assess the effect of PSA determination on mortality. In addition, we aimed to evaluate the PSA false-positive and -negative rate, and we did not consider the accuracy of PSA together with the determination of digital rectal examination or other genetic biomarkers (such as carriers of the Breast Cancer type 2 mutation). Given the high rate of false-positive results in the PSA determination, these factors should be considered for a final biopsy decision. Lastly, we collected the data from medical records, and thus, there are some missing data such as the Gleason classification, smoking habit, or the family history of PCa, among others.

## 5. Conclusions

In conclusion, this study showed a high rate of false-positive results in clinical practice for both symptomatic and asymptomatic patients, mainly in patients over 50 years. This study has generated relevant information related to the frequency and variables associated with the presence of false-positive results, which could be very useful for shared decision making. Although randomized control trials contribute to meaningful results, studies carried out in clinical practice are also needed to better support approaches to shared decision making.

## Figures and Tables

**Figure 1 cancers-15-00261-f001:**
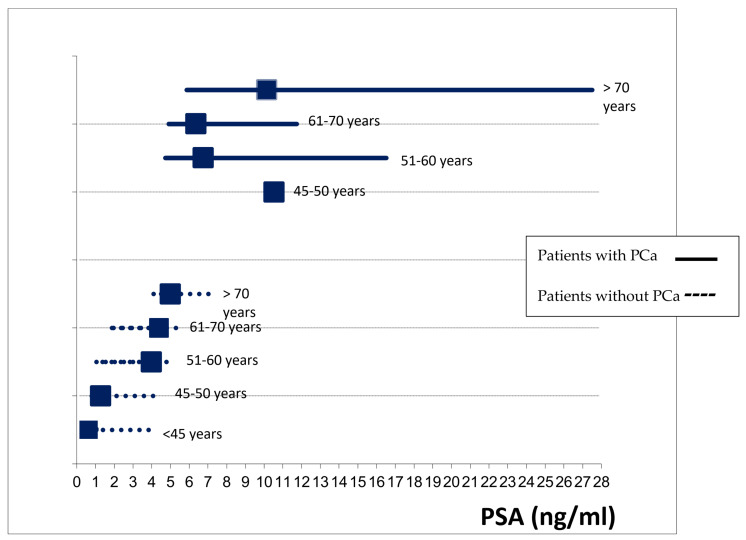
Comparison of PSA levels (median, IQR) between patients with and without PCa by age groups.

**Table 1 cancers-15-00261-t001:** Description of the sociodemographic and clinical characteristics associated with false-positive rates for both symptomatic and asymptomatic patients.

	Total			Symptomatic			Asymptomatic		
Variables	Total Cancer/Patients (109/1664; 6.5%)	False-Positive PSA Rate ^1^ (728/1555; 46.8%)	*p*-Value ^2^	Total Cancer/Patients (17/303; 5.6)	False-Positive PSA Rate ^1^ (137/286; 47.9)	*p*-Value ^2^	Total Cancer/Patients (92/1361; 6.8)	False-Positive PSA Rate ^1^ (591/1269; 46.6)	*p*-Value ^2^
Age			<0.001			0.070			<0.001
<45	0/57; 0	15/57; 26.3		0/19; 0	5/19; 26.3		0/38; 0	10/38; 26.3	
45–50	1/81; 1.2	22/80; 27.5		1/14; 7.1	6/13; 46.2		0/67; 0	16/67; 23.9	
51–60	12/369; 3.3	139/357; 38.9		3/59; 5.1	21/56; 37.5		9/310; 2.9	118/301; 39.2	
61–70	51/527; 9.7	229/476; 48.1		5/98; 5.1	52/93; 55.9		46/429; 10.7	177/383; 46.2	
>70	45/630; 7.1	323/585; 55.2		8/113; 7.1	53/105; 50.5		37/517; 7.2	270/480; 56.3	
PSA previous			0.008			0.147			0.023
No	29/315; 9.2	154/286; 53.8		6/55; 10.9	28/49; 57.1		23/260; 8.8	126/237; 53.2	
Yes	80/1347; 5.9	572/1267; 45.1		11/247; 4.5	108/236; 45.8		69/1100; 6.3	464/1031; 45	
Family history of PCa			0.003			0.899			0.002
No	19/169; 11.2	89/150; 59.3		3/34; 8.8	16/31; 51.6		16/135; 11.9	73/119; 61.3	
Yes	2/33; 6.1	17/31; 54.8		1/9; 11.1	4/8; 50		1/24; 4.2	13/23; 56.5	
Unknown	88/1462; 6	622/1374; 45.3		13/260; 5	117/247; 47.4		75/1202; 6.2	505/1127; 44.8	
Tobacco			0.274			0.056			0.677
Never smoker	21/377; 5.6	184/356; 51.7		4/69; 5.8	39/65; 60		17/308; 5.5	145/291; 49.8	
Current smoker	27/362; 7.5	154/335; 46		4/66; 6.1	24/62; 38.7		23/296; 7.8	130/273; 47.6	
Ex-smoker	40/546; 7.3	239/506; 47.2		6/103; 5.8	49/97; 50.5		34/44	190/409; 46.5	
Alcohol			0.014			0.836			0.004
Never drinker	22/386; 5.7	168/364; 46.2		5/75; 6.7	38/70; 54.3		17/311; 5.5	130/294; 44.2	
Current drinker	26/346; 7.5	181/320; 56.6		3/69; 4.3	36/66; 54.5		23/277; 8.3	145/254; 57.1	
Ex-drinker	3/56; 5.4	31/53; 58.5		0/7; 0	3/7; 42.9		3/49/6.1	28/46; 60.9	

^1^ False-positive rate: (False-positive results/patients without cancer) * 100; ^2^ *p*-value: Differences in the false-positive PSA rate.

**Table 2 cancers-15-00261-t002:** Description of the comorbidities associated with false-positive rates for both symptomatic and asymptomatic patients.

Variables	Total			With Symptoms			Without Symptoms		
	Total Cancer/Patients (109/1664; 6.5%)	False-Positive PSA Rate ^1^ (728/1555; 46.8%)	*p*-Value ^2^	Total Cancer/Patients (17/303; 5.6)	False-Positive PSA Rate ^1^ (137/286; 47.9)	*p*-Value ^2^	Total Cancer/Patients (92/1361; 6.8)	False-Positive PSA Rate ^1^ (591/1269; 46.6)	*p*-Value ^2^
Cardiovascular Disease			0.702			0.846			0.637
No	103/1508; 6.8	660/1405; 47		15/281; 5.3	127/266; 47.7		88/1227; 7.2	533/1139; 46.8	
Yes	6/156; 3.8	68/150; 45.3		2/22; 9.1	10/20; 50		4/134; 3	58/130; 44.6	
Hyperlipidemia			0.016			0.814			0.012
No	96/1391; 6.9	624/1295; 48.2		16/263; 6.1	119/247; 48.2		80/1128; 7.1	505/1048; 48.2	
Yes	13/273; 4.8	104/260; 40		1/40; 2.5	18/39; 46.2		12/233; 5.2	86/221;38.9	
Type II Diabetes Mellitus			0.002			0.008			0.019
No	98/1450; 6.8	654/1352; 48.4		15/275; 5.5	131/260; 50.4		83/1175; 7.1	523/1092; 47.9	
Yes	11/214; 5.1	74/203; 36.5		2/28; 7.1	6/26; 23.1		9/186; 4.8	68/177: 38.4	
Urinary tract infection			<0.001			0.007			<0.001
No	108/1617; 6.7	691/1509; 45.8		17/293; 5.8	128/46.4; 46.4		91/1324; 6.9	563/1233; 45.7	
Yes	1/47; 2.1	37/46; 80.4		0	9/10; 90		1/37; 2.7	28/36; 77.8	
IMC > 30			0.310			0.927			0.348
No	48/667; 7.2	291/619; 47		7/127; 5.5	52/120; 43.3		41/540; 7.6	239/499; 47.9	
Yes	20/333; 6	137/313; 43.8		2/56; 3.6	23/54; 42.6		18/277; 6.5	114/259; 44%	
HTA			0.175			0.590			0.212
No	18/352; 5.1	164/334; 49.1		2/70; 2.9	33/68; 48.5		16/282; 5.7	131/266; 49.2	
Yes	82/1154; 7.1	481/1072; 44.9		14/204; 6.9	85/190; 44.7		68/950; 7.2	396/882; 44.9	

^1^ False-positive rate: (False-positive results/patients without cancer) * 100; ^2^ *p*-value: Differences in the false-positive PSA rate.

**Table 3 cancers-15-00261-t003:** Description of the pharmacological treatments associated with false-positive rate for both symptomatic and asymptomatic patients.

	Total			Symptomatic			Asymptomatic		
Variables	Total Cancer/Patients (109/1664; 6.5%)	False-Positive PSA Rate ^1^ (728/1555; 46.8%)	*p*-value ^2^	Total Cancer/Patients (17/303; 5.6)	False-Positive PSA Rate ^1^ (137/286; 47.9)	*p*-value ^2^	Total Cancer/Patients (92/1361; 6.8)	False-Positive PSA Rate ^1^ (591/1269; 46.6)	*p*-Value ^2^
Treatment			0.003			0.611			0.003
No	1/334; 0.3	180/333; 54.1		0/59; 0	30/59; 50.8		1/275; 0.4	150/274; 54.7	
Yes	108/1330; 8.1	548/1222; 44.8		17/244; 7	107/227; 47.1		91/1086; 8.4	441/995; 44.3	
Statins			0.138			0.122			0.363
No	82/1201; 6.8	537/1119; 48		11/218; 5	105/207; 50.7		71/983; 7.2	432/912; 47.4	
Yes	27/463; 5.8	191/436; 43.8		6/85; 7.1	32/79; 40.5		21/378; 5.6	159/357; 44.5	
Metformin			0.415			0.059			0.979
No	101/1456; 6.1	629/1355; 46.4		16/267; 6	115/251; 45.8		85/1189; 7.1	514/1104; 46.6	
Yes	8/208; 3.8	99/200; 49.5		1/36; 2.8	22/35; 62.9		7/172; 4.1	77/165; 46.7	
Treatment for BPH			0.082			0.497			0.103
No	70/1305; 5.4	592/1235; 47.9		12/226; 5.3	105/214; 49.1		58/1079; 5.4	487/1021; 47.7	
Yes	39/359; 10.9	136/320; 42.5		5/77; 6.5	32/72; 44.4		34/282; 12.1	104/248; 41.9	
Diuretic			0.245			0.801			0.157
No	96/1482; 6.5	656/47.3		16/270; 5.9	121/254; 47.6		80/1212; 6.6	535/1132; 47.3	
Yes	13/182; 7.1	72/169; 42.6		1/33/3	16/32; 50		12/149; 8.1	56/137; 40.9	
ASA			0.234			0.481			0.329
No	95/1412; 6.7	625/1317; 47.5		11/266; 4.1	124/255; 48.6		84/1146; 7.3	501/1062; 47.2	
Yes	14/252; 5.6	103/238; 43.3		6/37; 16.2	13/31; 41.9		8/215; 3.7	90/207; 43.5	

^1^ False-positive rate: (False-positive results/patients without cancer) * 100; ^2^ *p*-value: Differences in the false-positive PSA rate.

**Table 4 cancers-15-00261-t004:** Description of the PSA levels (ng/mL) in those patients with PCa according to Gleason classification.

	Gleason			*p*-Value
PSA Levels	6	7	8–10	
PSA (ng/mL) (median, IQR)	6.66 (4.91–10.13)	6.32 (5.15–8.89)	13.37 (6.44–31.24)	0.004

**Table 5 cancers-15-00261-t005:** Differences in the PSA levels (ng/mL) according to the presence of some comorbidities and prescribed pharmacological treatment for patients without and with PCa.

	Total	Without PCa		With PCa	
Variables	PSA Levels (ng/mL) (Median, IQR)	PSA Levels (ng/mL) (Median, IQR)	*p*-Value	PSA Levels (ng/mL) (Median, IQR)	*p*-Value
Hyperlipidemia			0.009		0.082
• No	4.53 (1.89–6.36)	4.44 (1.67–6.01)		8.20 (5.18–16.84)	
• Yes	4.27 (1.24–5.15)	4.24 (1.18–5.13)		5.87 (4.44–6.61)	
Diabetes			0.001		0.974
• No	4.53 (1.90–6.26)	4.45 (1.73–5.91)		6.82 (5.08–16.80)	
• Yes	4.19 (1–5.39)	4.04 (0.92–5.08)		7.98 (5.93–14.45)	
UTI			<0.001		0.993
• No	4.45 (1.69–6.07)	4.36 (1.15–5.77)		6.86 (5.09–16.54)	
• Yes	5.71 (4.64–10.33)	5.70 (4.64–9.51)		10.33	
Pharmacological treatment			<0.001		0.484
• No	4.70 (2.29–6.35)	4.70 (2.29–6.34)		5.68 (5.08–6.28)	
• Yes	4.39 (1.64–6.07)	4.32 (1.41–5.74)		7.17 (5.11–16.77)	

## Data Availability

The data presented in this study are available on request from the corresponding author.

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
