# Peer review of "Variables Associated with False-Positive PSA Results: A Cohort Study with Real-World Data"

_cancers, 2022, doi:10.3390/cancers15010261_

Round 1

Reviewer 1 Report

The article assess the diagnostic performance of PSA biomarker in human samples. The introduction flow was informative with logic flow. The methods are well described in sufficient details. However, I have the following suggestions:

Since we are analyzing and comparing the efficiency of a diagnostic test in clinical context. Details on testing PSA methods is warranted. Are all cohorts underwent same technique, assay, lab, sample type and processing,...etc. What are pre-analytical, analytical and post-analytical measures that was set to unify the milestone of lab testing.

Line 156: why previous PSA testing was added to the model? What is its significance.

Results:

How the results of the study will change the management paradigm? e.g more FP in older age, while Pca is more in advanced age. What is the hypothesis driven from including the family history and being significantly different in Table 1.

Type in table 2 in hyperlipidemia line.

Does table 2 results means we should not use the test in the absence of DM and hyperlipidemia?

Table 3: since each treatment type did not show significance, it is confusing to use the overall treatment pooling them together which showed contradictory results.

Did you try different cutoff values for positive and negative PSA to define the optimum threshold in the presence of different biological variations and comorbidities.

Author Response

Response to Reviewer 1 Comments

The article assesses the diagnostic performance of PSA biomarker in human samples. The introduction flow was informative with logic flow. The methods are well described in sufficient details. However, I have the following suggestions:

Since we are analyzing and comparing the efficiency of a diagnostic test in clinical context. Details on testing PSA methods is warranted. Are all cohorts underwent same technique, assay, lab, sample type and processing, etc. What are pre-analytical, analytical and post-analytical measures that was set to unify the milestone of lab testing.

We have added in the methods section a paragraph detailing the procedure undergone to determine PSA levels (methods section page 3, lines 122-126).

Line 156: why previous PSA testing was added to the model? What is its significance.

It was a mistake; we have deleted this variable from the statistical analysis section.

Results:

How the results of the study will change the management paradigm? e.g more FP in older age, while Pca is more in advanced age.

Regarding the management paradigm the results of our study support restricting screening in older men. It is true that PCa is more frequent in advanced age but the level of PSA in the blood of health men also increases with age by about 3.2% per year (Oesterling JE, Jacobsen SJ, Chute CG, Guess HA, Girman CJ, Panser LA, et al. Serum prostate-specific antigen in a community-based population of healthy men: establishment of age-specific reference ranges. JAMA J Am Med Assoc 1993;270:860–4). We show here that proportion of false positive PSA results is high in this group. At the same time, prostate cancer overdiagnosis (cases that would not have caused clinical consequences during a man’s lifetime if left untreated) has a strong relationship to age (Vickers AJ, Sjoberg DD, Ulmert D, Vertosick E, Roobol MJ, Thompson I, Heijnsdijk EA, De Koning H, Atoria-Swartz C, Scardino PT, Lilja H. Empirical estimates of prostate cancer overdiagnosis by age and prostate-specific antigen. BMC Med, 2014, 12, 26). Therefore, a relevant percentage of the prostate cancer cases detected in asymptomatic patients included in the study could also be overdiagnosed cases. In this way, our results strengthen the evidence base in support of available guidelines which recommend restricting screening in patients older than 70 years old.

We have added this explanation in the discussion section (page 14, lines 305-320).

What is the hypothesis driven from including the family history and being significantly different in Table 1.

A family history of prostate cancer may increase the risk of prostate cancer in the population undergoing screening. The hypothesis here would be that the predictive value of a positive PSA test in a person with a family history of prostate cancer may be higher than that of a person without a family history of prostate cancer. However, we have a limitation here that this data was not available in the clinical history in 1462 (87.9%) patients, so we have a large number of missing data. For this reason, we are happy to we have eliminated this variable from the table 1 and we have described it in the text (result section, page 4, lines 180-183).

Type in table 2 in hyperlipidemia line.

Done

Does table 2 results means we should not use the test in the absence of DM and hyperlipidemia?

The results on DM and hyperlipidemia are curious but rather that suggesting not using the test in these populations they suggest that a positive PSA screening test is more suggestive of cancer (lower false positive rate, higher positive predictive value for cancer). This can be explained by the fact that patients with hyperlipidemia and diabetes have lower PSA levels than those without these pathologies. A recent systematic review showed that diabetes was significantly associated with lower PSA levels among asymptomatic men older than 60 years (Bernal-Soriano MC, Lumbreras B, Hernández-Aguado I, Pastor-Valero M, López-Garrigos M, Parker LA. Untangling the association between prostate-specific antigen and diabetes: a systematic review and meta-analysis. Clin Chem Lab Med, 2020, 59, 11-26). Although lower range of PSA values identified in diabetics compared to non-diabetics could justify a different cut-off point when considering a positive PSA result in this population, it is unlikely that using a diabetes specific cut-off would improve accuracy of PSA screening.

We have added this explanation in the discussion section (page 14, lines 328-338).

Table 3: since each treatment type did not show significance, it is confusing to use the overall treatment pooling them together which showed contradictory results.

The presence of any prescribed treatment in the patients included in the study was collected as well as data on specific treatments that had previously shown an impact on PSA levels such as statins, metformin, BPH treatment, diuretics and ASA. Thus, while each specific type of treatment did not show a significant association with the presence of false positive results, having a pharmacological prescription (including any pharmacological treatment) did show a significant association in the univariate analysis. We don’t feel this is necessarily contradictory to the absence of statistical significance when analysing treatment type, which could perhaps by explained by the limited statistical power. Moreover, most of the patients included in this study had two or more concomitant treatments, and it was difficult to distinguish the specific treatment responsible for the increased likelihood of having a false positive result in the univariate analysis. We feel its relevant to share the finding that pharmacological treatments can influence false positive rates as it complements the existing evidence from other studies about the interaction between pharmacological treatments and PSA levels.

However, in line with the review comments we have addressed these issues in further detail in the manuscript. We have included a description of this variable in the method section (page 3, lines 140-141) and in the discussion section (page 15, lines 374-383).

Did you try different cutoff values for positive and negative PSA to define the optimum threshold in the presence of different biological variations and comorbidities.

PSA values were different from patients without prostate cancer according to age, diabetes, hyperlipidemia, urinary tract infection and those with a pharmacological treatment. However, given that there were no differences in PSA values among patients with PCa, it is unlikely that using a specific cut-off would improve accuracy of PSA screening.

 We have included this explanation in the discussion section (page 14, lines 333-338).

Reviewer 2 Report

Authors aimed to evaluate the incidence of false positive and false negative results in a general clinical setting, including patients undergoing opportunistic screening or with symptoms suggestive of disease, and the variables associated. Patients were followed by reviewing their medical records for 2 years or until the diagnosis of PCa was reached, whichever came first.

False positive PSA rate was 46.8% and false negative PSA rate was 2.8%. Patients aged 61-70 years and those over 70 years were more likely to have a false positive result than those under 45 years. Patients with urinary tract infection were more likely to have a false 39 positive result. Patients with diabetes mellitus were less likely 40 to have a false positive result.

The issue addressed is of clinical relevance and methodology is correct.

Results: Of the 109 patients with PCa, 86.2% were able to be classified according to the Gleason score: (<6, 6-7, 8-10).  Any information about their International Society of Urological Pathology grade group? Please, comment about the unusual finding of Gleason score <6.

Moreover, I suggest to improve the discussion section by discussing the following issues:

1.     Role of mpMRI, PSA derivatives, molecular tests, and their combination in patients with elevate PSA (See for example: Cuocolo R, Stanzione A, Rusconi G, Petretta M, Ponsiglione A, Fusco F, Longo N, Persico F, Cocozza S, Brunetti A, Imbriaco M. PSA-density does not improve bi-parametric prostate MR detection of prostate cancer in a biopsy naïve patient population. Eur J Radiol. 2018 Jul;104:64-70. doi: 10.1016/j.ejrad.2018.05.004. Epub 2018 May 4. PMID: 29857868.).

2.     PSA alterations due to finasterise/dutasteride.

3.     Interestingly, the presence of diabetes  mellitus type II was associated with a lower probability of having a false positive  result: the rationale behind this finding deserves more accurate discussion.

4.     I also suggest to discuss findings from PSA screening adoption and knowledge in other geographic areas (see for example: Mirone V, Imbimbo C, Arcaniolo D, Franco M, La Rocca R, Venturino L, Spirito L, Creta M, Verze P. Knowledge, attitudes, and practices towards prostate cancer screening amongst men living in the southern Italian peninsula: the Prevention and Research in Oncology (PRO) non-profit Foundation experience. World J Urol. 2017 Dec;35(12):1857-1862. doi: 10.1007/s00345-017-2074-9. Epub 2017 Aug 5. PMID: 28780740.).

Author Response

Response to Reviewer 2 Comments

Authors aimed to evaluate the incidence of false positive and false negative results in a general clinical setting, including patients undergoing opportunistic screening or with symptoms suggestive of disease, and the variables associated. Patients were followed by reviewing their medical records for 2 years or until the diagnosis of PCa was reached, whichever came first. False positive PSA rate was 46.8% and false negative PSA rate was 2.8%. Patients aged 61-70 years and those over 70 years were more likely to have a false positive result than those under 45 years. Patients with urinary tract infection were more likely to have a false 39 positive result. Patients with diabetes mellitus were less likely 40 to have a false positive result. The issue addressed is of clinical relevance and methodology is correct.

Results: Of the 109 patients with PCa, 86.2% were able to be classified according to the Gleason score: (<6, 6-7, 8-10).  Any information about their International Society of Urological Pathology grade group? Please, comment about the unusual finding of Gleason score <6.

Thank you for your comments. We also collected information about the International Society of Urological Pathology grade in 43 patients. We have included this information in the result section (page 12, lines 249-253).

Regarding the unusual finding of Gleason score <6, this was a mistake. The patients were classified according to the Gleason score as 6, 7, 8-10. We have changed the data in the result section (page 12, lines 242-245).

Moreover, I suggest improving the discussion section by discussing the following issues:

  1. Role of mpMRI, PSA derivatives, molecular tests, and their combination in patients with elevate PSA (See for example: Cuocolo R, Stanzione A, Rusconi G, Petretta M, Ponsiglione A, Fusco F, Longo N, Persico F, Cocozza S, Brunetti A, Imbriaco M. PSA-density does not improve bi-parametric prostate MR detection of prostate cancer in a biopsy naïve patient population. Eur J Radiol. 2018 Jul;104:64-70. doi: 10.1016/j.ejrad.2018.05.004. Epub 2018 May 4. PMID: 29857868.).

In accordance with reviewers’ advice, we have included a paragraph discussing these aspects in the discussion section (pages 15-16, lines 398-419).

  1. PSA alterations due to finasterise/dutasteride.

We have included a paragraph discussing the effect of finasteride/dutasteride in the reduction of PSA levels (pages 15 lines 374-380).

  1. Interestingly, the presence of diabetesmellitus type II was associated with a lower probability of having a false positive  result: the rationale behind this finding deserves more accurate discussion.

We have included a discussion of this topic in the discussion section (page 14, lines 322-338).

  1. I also suggest discussing findings from PSA screening adoption and knowledge in other geographic areas (see for example: Mirone V, Imbimbo C, Arcaniolo D, Franco M, La Rocca R, Venturino L, Spirito L, Creta M, Verze P. Knowledge, attitudes, and practices towards prostate cancer screening amongst men living in the southern Italian peninsula: the Prevention and Research in Oncology (PRO) non-profit Foundation experience. World J Urol. 2017 Dec;35(12):1857-1862. doi: 10.1007/s00345-017-2074-9. Epub 2017 Aug 5. PMID: 28780740.).

We have included a discussion about the PSA knowledge in other countries, considering the new European recommendations on shared-decision making in the discussion section (page 16, lines 420-428).
